# The Polymorphism Asn680Ser on the FSH Receptor and Abnormal Ovarian Response in Patients with Normal Values of AMH and AFC

**DOI:** 10.3390/ijms24021080

**Published:** 2023-01-05

**Authors:** Giorgio Maria Baldini, Assunta Catino, Simone Palini, Romualdo Sciorio, Daniele Ferri, Marina Vinciguerra, Domenico Baldini

**Affiliations:** 1IVF Center, Momo Fertilife, 76011 Bisceglie, Italy; 2Department of IVF, “San Giorgio” Hospital—AUSL Romagna, 47841 Cervia, Italy; 3Edinburgh Fertility Reproductive Center, Royal Infirmary Edinburgh, Edinburgh EH16 4SA, UK; 4Department of Biomedical Sciences and Human Oncology, Obstetrics and Gynaecology Section, Univesity of Bari, 70121 Bari, Italy; 5Clinic of Obstetrics and Gynecology “Santa Caterina Novella”, Hospital Galatina, 73013 Galatina, Italy

**Keywords:** Asn680Ser FSH receptor, oocytes retrieval, controlled ovarian stimulation, AMH and AFC

## Abstract

After the controlled ovarian stimulation (COS), the number of cumulus oocyte complexes collected is lower than predicted. The aim of this study is to understand if there is a possible reason for that deficient ovarian response. It was hypothesized that this is associated with the SNP (single-nucleotide polymorphism) of the FSH receptor (FSHr), specifically c.2039A > G, resulting in Asn680Ser. Two groups of patients were enrolled for this purpose: the normal (*n* = 36) and abnormal responses (*n* = 31). To predict the number of retrievable oocytes, according to the anti-Mũllerian hormone (AMH) and the antral follicle count (AFC), the following formula was applied in a log scale: the number of oocytes retrieved = 2.584 − 0.015 × (age) − 0.035 × (FSH) + 0.038 × (AMH) + 0.026 × (AFC). Then, when the number of oocytes collected was less than 50% of the calculated value, it was proposed that the patients result in an abnormal response. DNA sample blood was collected from the women, and then the genetic assessment for the Asn680Ser of the FSHr was evaluated in both groups. The differences between the two categories were statistically analyzed with an independent samples *t* test, a Mann–Whitney U test and a Chi-squared test. In a patient with an abnormal response, a significant prevalence of the amino acid serine at position 680 of the FSHr compared to the counterpart group (*p* < 0.05) was detected. In conclusion, according to the results, the genetic evaluation of the FSHr could represent an accurate and predictive feature for patients undergoing assisted reproductive technology treatment.

## 1. Introduction

In assisted reproductive technology, the concentrations of the serum AMH and AFC values have emerged as the preferred methods for assessing the ovarian reserve [1,2,3,4]. They have become indeed the most applied markers to predict the response to controlled ovarian stimulation [5,6]. An AMH between 0.1 and 1.26 ng/mL is indicative of patients with a reduced ovarian reserve, with a sensitivity of 80–87% and a specificity of 64–93% [7]. Several studies have shown a positive correlation between high levels of the AMH (serum and follicular) and the numbers of mature (MII) retrieved oocytes, the fertilization rate, as well as the embryo development rate and pregnancy outcome [8,9,10]. The AFC consists of a high-resolution ultrasound evaluation of follicles with dimensions between 2 and 10 mm [11], assessed in the first days of the menstrual cycle. The AFC is considered normal if at least 10 ± 4 antral follicles are detected [12,13]. The literature seems to confirm that both the AMH and AFC are equally effective in predicting a poor or an excessive ovarian response, showing overlapping accuracy [14,15,16,17]. 

Moon et al., in 2016, published an article focusing on how to predict the number of retrievable oocytes, according to the AMH levels and AFC number [18]. Despite the validity of the AMH and AFC, a small percentage of healthy patients undergoing a COS with optimal AMH and AFC values showed a poor response when compared to the predictions. In the literature, nothing was found that could explain this kind of response in these women and hence it was hypothesized that the cause could be genetic. In the Delphi consensus [19], a link between some variants in the gonadotropin receptor genes and ovarian stimulation outcomes is supported. The most studied is the FSH receptor because the follicle stimulating hormone is essential for ovarian development and function [19]. In women, the FSH stimulates the growth and maturation of the antral follicles in the ovaries and promotes ovulation [20]. Further, it increases the estradiol production through binding to the FSH receptor, expressed by granulosa cells. The FSHr belongs to the family of transmembrane G protein-coupled receptors. The N-terminal domain comprising 349 amino acids is located in the extracellular portion and is responsible for binding the FSH. The transmembrane domain is composed of 264 amino acids and has a structure that assures the stability of the receptor [20]. The intracellular C-terminal domain contains 65 amino acids; it comprises several phosphorylation sites that are able to trigger the transduction signal following binding to the FSH [21]. The phosphorylation of the intracellular domain activates the adenylate cyclase system, inducing an increase in the intracellular cyclic AMP level. Furthermore, the receptor triggers the mitogen-activated protein-kinase (MAPK) pathway, with the synthesis of intracellular calcium and inositol triphosphate (IP3) [22]. Finally, the gene that codes for the FSHr is about 190 Kb long and is located on chromosome 2 in position p21-p16; its coding region includes 10 exons and 9 introns [23].

Various single-nucleotide polymorphisms were identified in the FSHr coding region, and the most studied types are Thr307Ala (rs6165) and Asn680Ser (rs6166) [19,20,21,24]. The presence of both polymorphisms on exon 10 causes a strong linkage disequilibrium (non-random combination of alleles) between them [25]. The two polymorphisms are located in the same exonic region, and it means that combinations of p.Thr307-p.Asn680 and p.Ala307-p.Ser680 are inherited together; therefore, they bind to each other during recombination and do not show random distribution [25]. Hence, we have focused only on the Asn680Ser polymorphism in this study. The Ans680Ser polymorphism affects the intracellular domain of the receptor, where an asparagine residue at position 680 is replaced by a serine residue. This SNP might induce adverse effects on the functionality of the FSHr, involving the female reproductive outcome [21]. 

Based on this information, it was supposed that the polymorphism Asn680Ser on the FSHr human female gene could play a role in the reductive response to the exogenous gonadotropin administered during the COS, leading to a poorer eggs retrieval than predicted. The objective of this study was to clarify a possible reason for a deficient ovarian response in healthy patients with optimal AMH levels and AFC values; as a consequence, we suggest performing a genetic analysis of Asn680Ser to predict the number of recoverable oocytes after the COS and customize the stimulation protocol for these patients who are looking for a pregnancy.

## 2. Results

The patients were divided into two groups according to the response at the COS: the abnormal responders (31/67) and normal responders (36/67). The genetic analysis of Asn680Ser on the FSHr revealed that in the abnormal group, 23% (7/31) of the patients were homozygous for Asn, 54% (17/31) heterozygous and 23% (7/31) homozygous for Ser. In the group of ‘normal responders’, 64% (23/36) of the women were homozygous for Asn, 25% heterozygous and 11% (4/36) homozygous for Ser. In the abnormal responders group, there was a significant prevalence of the amino acid Ser at position 680 of the FSHr compared to the normal responders (*p* < 0.01). Further, in the normal responders group, the majority of the patients showed to be homozygous for asparagine at position 680 of the FSHr (Figure 1). Finally, the percentage of fertilized oocytes was significantly higher in the normal responder group (*p* < 0.05) (Table 1). However, no differences were observed on both numbers of mature oocytes collected after pick-up, embryo obtained and embryo transferred between the two groups (*p* ≥ 0.05; Table 1).

## 3. Discussion

The results obtained show that in healthy patients undergoing a COS with an abnormal ovarian response, there is a prevalence of amino acid serine at position 680 on the FSHr gene, compared to normal responders where the asparagine in the same position prevails (*p* < 0.05). In addition, a statistically significant difference (*p* < 0.05) in terms of the fertilization rate between the normal and abnormal responders patients following a COS is found (Table 1). 

The abnormal response obtained corresponds to a poor oocytes recovery compared to that expected according to the AMH and AFC. This event could be explained by the role of the FSHr during the COS: the administration of the FSH exogenous induces an increase in the estradiol through binding to the FSH receptor. Different authors have demonstrated that the Ser680 causes a partial resistance of the receptor to gonadotropin; therefore, the basal levels of the FSH will be higher in these patients than in those with Asn680 [25,26,27]. Indeed, considering the higher dose of exogenous FSH, women with a homozygous Ser680 genotype show lower estradiol (E2) levels and require a higher dose of exogenous FSH to achieve the same E2 levels as women who are homozygous for Asn680 or heterozygous [28,29,30,31]. These findings agree with a study that found that the same dose of FSH correlates with lower levels of E2 in patients homozygous for Ser680 when compared with women homozygous for Asn680 [26]. On the other hand, an increased sensitivity to the exogenous FSH in patients carrying the Asn680 variant (particularly if homozygous) was observed. Therefore, these women might be exposed to a higher risk of developing ovarian hyper-stimulation syndrome (OHSS) compared to patients carrying the Ser680 allele [32,33]. Indeed, patients homozygous for Asn680 are particularly reactive to the FSH, and following a COS, they produce a relevant number of follicles, as well as high levels of E2 in the circulation [34]. Instead, the heterozygous patients are those who respond in a more active, balanced way to controlled ovarian stimulation [35].

In this study, a statistical significance (*p* < 0.05) in terms of the fertilization rate between the two groups was found: a higher fertilization rate was obtained from the normal group, probably due to the presence of at least one Ser in position 680 in the FSHr. Further studies need to be performed to clarify this result. Moreover, in the literature, no information was recovered.

Regarding the rate of mature oocytes following pick-up, the results obtained show that there is not a significant difference between the two groups. However, in the literature, it was demonstrated that women carrying the Ser680 variant (homozygous specially) yield a lower number of mature oocytes recovered compared with patients who have at least one Asn680 residue [36]. Nevertheless, other authors found that in patients homozygous for Asn680, the number of mature oocytes is reduced compared to homozygotes for Ser680 [37]. Although, the absence of a relationship between the FSHr Asn680Ser genotype and the number of recoverable metaphase II oocytes was also reported [38,39]. The debate in the literature on this concern is still very active, and it is currently not possible to draw a firm conclusion. 

According to the results obtained, it could be possible that the presence of Ser680 or Asn680 in the FSHr gene does not affect the values of both the AMH and AFC. Indeed, it was demonstrated that there is no relationship between the Asn680Ser polymorphism of the FSHr and AMH levels [40]. To predict the ovarian response following a COS, it is recommendable to determinate the genotype of the FSHr, especially when the patients are healthy and with an optimal AMH level and AFC value. These findings support the initial hypothesis that despite the optimal values of the AMH and AFC, the cause of an abnormal oocyte response may be related to the amino acid substitution in position 680 of the FSH receptor. 

One of the main challenges of this study was understanding how to determine when the ovarian response was abnormal. For this purpose, the most recent literature was analyzed and only one article investigated the relationship between the prediction of the number of oocytes retrieved after a COS and the ovarian reserve biomarkers [18]. In addition, there is a lack of data in the literature exploring the correlation between the oocyte number predicted and obtained. For this reason, we set this value as the following: when the oocyte retrieval number is less than 50% of the predicted value, the responses were considered poor. Furthermore, when the formula suggested it was applied, we noticed abnormal results in the patients with very high levels of AMH and AFC values. On the other hand, when the AMH and AFC are deficient, the clinical effects could be already in an advanced stage. This condition could expose the patients to the premature depletion of the ovarian reserve. Nevertheless, further investigation which includes a higher number of patients is needed to confirm the results. Our study represents a novelty because there is still little literature, and the opinions are controversial. This is not a randomized controlled trial; it is a comparative and retrospective study. Therefore, our findings should be validated with a prospective trial.

In conclusion, despite that the AMH and AFC are the most reliable ovarian biomarkers, occasionally they might be not able to predict an accurate response after a COS. Probably the ovarian response in terms of the oocytes collected depends on the genotype of the FSH receptor, and in this study, it was focused on only one polymorphism: Asn680Ser (the most studied and cited from the literature). The polymorphism does not affect the levels of the AMH and AFC, but additional studies are needed to clarify the details of those mechanisms and to help understand the reason why some patients respond to a COS differently than the prediction. Moreover, to increase the pregnancy outcomes for infertile patients, a tool to establish when a COS results in an abnormal oocyte response will allow clinicians to customize the stimulation protocol. 

## 4. Materials and Methods

### 4.1. Patients

This is a case-control study conducted at the IVF MOMO’ FertiLife Centre (Bisceglie, Italy), from January 2012 to December 2021 (Figure 2). During this time, 4987 ART treatments were performed using the intracytoplasmic sperm injection (ICSI) as insemination method. Out of 4987 treatments, 31 patients were retrospectively selected and allocated to the ‘abnormal responder’ group, who, despite normal AMH and AFC values, showed a low number of MII oocytes retrieved. 

The Asn680Ser genotypic analysis of FSHr was examined in this group of selected patients. It was compared to the control group of ‘normal responder’, including 36 patients with same characteristics of age, BMI or cause of infertility and showing a number of oocytes retrieved concordant with the AMH and AFC values. The patients enrolled in the ‘normal responders’ group were randomly selected through the MedITEX software version number 2.8.7.4. All patients recruited in the study were heathy, with regular menstrual cycle of 25–35 days, age ≤ 45 years and at least 1 year of infertility. The patients with the characteristics reported in Table 2 were excluded. The receptor polymorphism analysis was also investigated in the control group.

### 4.2. Ethics Approval and Consent to Participate

The Local Ethics Committee (LEC) of the MOMO’ FERTILIFE Center believes that the above research, for the purpose of the study, teaching and training indicated in the project, is ethically justified, recommending that the following conditions to be documented and guaranteed:That couples signed the informed consent for the purpose study, research and training;That the informed consent has been expressed by an adult, aware and in the absence of any form of material or psychological coercion.

The opinion in response to the question on the ethical aspects concerning the above research was discussed in the LEC, session of 18 December 2017, and approved unanimously (prot. N. 0718).

### 4.3. Stimulation and Prediction of the Ovarian Response 

All patients undergoing ART treatment performed the AMH test to assess the ovarian reserve. The dosage was performed with a chemiluminescence immunometric system (MAGLUMI 800, Snibe, Shenzhen, China). The AFC evaluation was completed on the first day of the cycle with a GE Voluson S8 ultrasound system (GE Healthcare, Chicago, IL, USA). The examination was always performed by the same doctor. Thereafter, the patients underwent COS using the GnRH-antagonist protocol. The recombinant follicle-stimulating hormone (FSH; GONAL-f, Merck Serono, Weiterstadt, Germany) was administered from the second day of menstruation; the starting dose was determined always using La Marca nomogram [49]. Follicular maturation was monitored through serum estradiol (E2) and progesterone (P) levels and in parallel, also with transvaginal follicular ultrasound, respectively, on days 5, 7, 10 and 12 from the start of therapy. When one or more follicles reaching 13–14 mm size, 0.25 mg of Cetrotide (Merck Serono, Weiterstadt, Germany) was administered daily to avoid ovulation. Upon reaching at least 3 follicles with a size ≥ 18 mm in diameter, a dose of 10,000 IU of chorionic gonadotropin (hCG) was administered to induce ovulation (GONASI, IBSA, Lugano, Switzerland). The prediction of the ovarian response was calculated using the equation (log scale) stated above: (number of oocytes retrieved) = 2.584 − 0.015 × (age) − 0.035 × (FSH) + 0.038 × (AMH) + 0.026 × (AFC) [18]. At the time of the oocyte retrieval, if the number of MII oocytes recovered was 50% lower than the value calculated by the equation, we then classified the ovarian response as ‘deficient’ (Table 3). 

### 4.4. Hormonal Assays

Serum AMH was measured using a specific immunometric assay MAGLUMI 800 kit (Sinbe Diagnostic, Shenzhen, China). The sensitivity and intra/inter-assay coefficients of variation (CVs) for AMH assay were 0.08 ng/mL and 4.2/5.7%, respectively. E2 and progesterone were also measured by competitive immunoassay using the MAGLUMI 800 kit (Sinbe Diagnostic, Shenzhen, China), with intra/inter-assay CVs of 4.1/5.6 and 3.9/5.7% for E2 and progesterone, respectively. The sensitivity for E2 and progesterone was 2 and 4.7 pg/mL, respectively.

### 4.5. Oocyte Pick-Up, ICSI and ET

The ultrasound-guided transvaginal oocyte retrieval was performed (OPU; VOLUSON S8, GE Healthcare, Chicago, IL, USA) [45]. For each patient, the total number of mature oocytes retrieved was assessed using a stereo microscope (Nikon SMZ 1500, Tokyo, Japan). With the use of an inverted microscope and a micromanipulator, the mature MII oocytes were injected by an embryologist with the ICSI technique selecting the morphologically normal spermatozoa through the horizontal migration technique [42,50]. The embryos were cultured in a single-step media (G-TLTM, Vitrolife, Gothenburg, Sweden) using a time lapse incubator (GERITM, Genea Biomedx, Sydney, Australia), adopting a pre-mixed gas with 5% O_2_ and 6% CO_2_ and balance nitrogen. After 16–19 h post injection, fertilization was assessed with the presence of two pronuclei and the second polar body in the perivitelline space. Further embryonic development was monitored daily, and on day 5 at blastocyst stage, the most competent embryo selected by morphological evaluation was transferred to the uterine cavity [51]. The β-hCG dosage to determine the outcome of the pregnancy was completed 12 days after the embryo transfer (ET). Subsequently, the viable pregnancy was confirmed by transvaginal ultrasound.

### 4.6. DNA Extraction and PCR-RFLP Analysis for the Asn680Ser Variant

A blood sample of 1 mL was collected from each patient, using a collection tube with EDTA as anticoagulant. Genomic DNA was obtained from peripheral blood leukocytes with the WizardTM Genomic DNA Purification Kit (Promega, Madison, WI, USA) according to the manufacturer’s instructions. The presence of the Asn680Ser variant introduces a restriction site that can be used for the PCR–RFLP (PCR–restriction fragment length polymorphism) technique. The region of nucleotide number 1624 to 2143 in the FSHr gene was amplified by PCR with Applied Biosystems SimpliAmp Thermal Cycler (Thermo Fisher, Waltham, MA, USA) using genomic DNA as templates and a set of primers [29] (primer-1: 5′-TTTGTGGTCATCTGTGGCTGC-3′; and primer-2: 5′-CAAAGGCAAGGACTGAATTATCATT-3′) which amplified the DNA fragment of 520 bp in size. Following, the PCR fragment was cut by restriction enzyme (BseNI) approach. Standard electrophoresis in 2% agarose gel and 0.5× Tris-borate EDTA buffer was used to separate the uncut PCR product (520 bp) representing the A allele (Asn) from the restricted PCR fragments (413 and 107 bp) corresponding to the G allele (Ser).

### 4.7. Statistical Analysis

The whole sample was categorized according to the ovarian response after COS, as described in Methods section (abnormal response as compared to the normal response group). The Shapiro–Wilk test was performed to assess the normality of variables in each group. Independent samples t test, Mann–Whitney U test and Chi-squared test were performed to assess differences between groups, for continuous and categorical variables, respectively. The statistical analysis was performed with R Software Version 1.4.1106. 

## Figures and Tables

**Figure 1 ijms-24-01080-f001:**
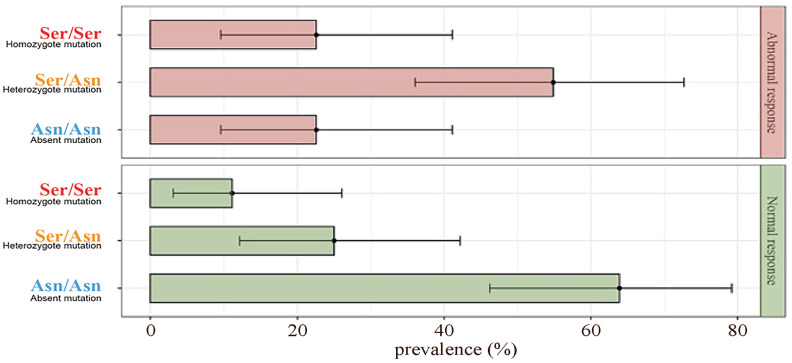
Prevalence bar plot of FSHr mutation within response groups.

**Figure 2 ijms-24-01080-f002:**
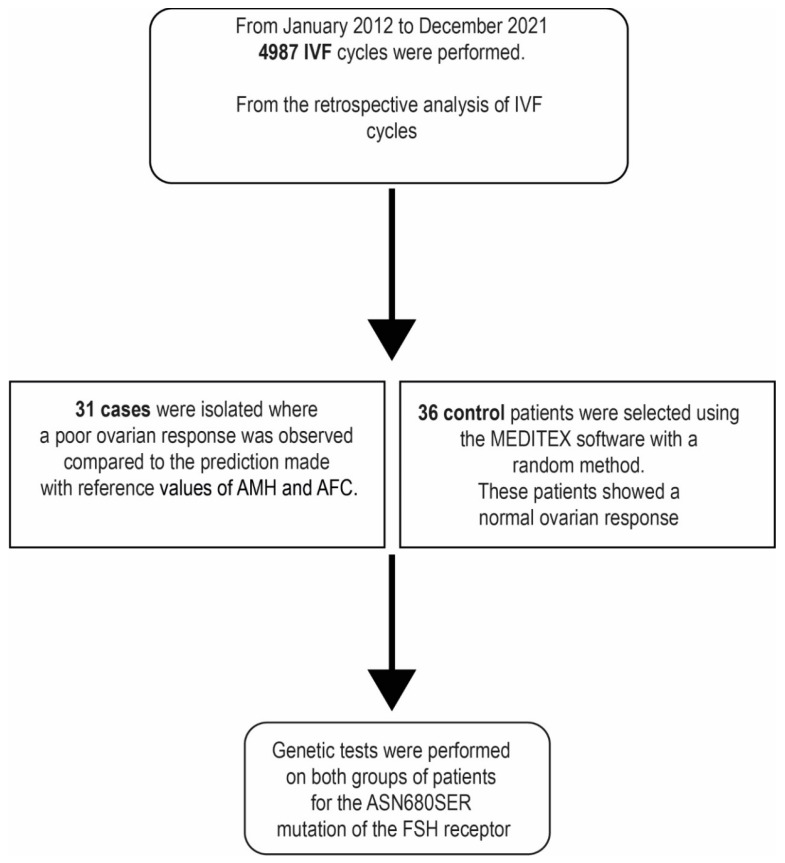
Study design.

**Table 1 ijms-24-01080-t001:** Patients’ (*n* = 67) results of clinical, biochemical and ICSI cycles. All data are shown as average (±) and SD and medians (from min. to max.) or continuous variables and as n° and proportions (%). Mann–Whitney U test was not otherwise specified, independent sample *t* test and χ^2^ Chi-squared test. ^1^ BMI = Body Mass Index; ^2^ LH = Luteinizing Hormone. The underlines highlight the statistical significance.

	Abnormal Response		Normal Response		*p* Value
Proportions (%)	31 (46.30)		36 (53.70)		
Age (years)	35.16 ± 3.75	35 (28 to 42)	35.25 ± 3.75	35.50 (28 to 42)	0.92
BMI (Kg/m^2^) ^1^	22.79 ± 3.72	21.80 (17.40 to 34.10)	22.43 ± 3.14	22.20 (16 to 29.40)	0.93
AMH (ng/mL)	3.25 ± 3.13	2.59 (0.84 to 17.60)	3.10 ± 3.75	1.89 (0.24 to 18.09)	0.11
Total FSH (IU)	2464.96 ± 1175.40	2100 (1109 to 5100)	2342.32 ± 1139.76	2211 (721.50 to 5100)	0.26
FSH (mIU/mL)	8.01 ± 3.23	7.60 (1.76 to 15.7)	8.24 ± 4.97	7.64 (0.35 to 26.28)	0.72
LH (mIU/mL) ^2^	5.57 ± 2.56	4.84 (2.29 to 12.13)	6.08 ± 2.74	5.80 (0.83 to 11.81)	0.30
AFC (*n*)	12.7 ± 5.31	13 (3 to 26)	12.30 ± 5.69	11.50 (3 to 25)	0.76
E2 (pg/mL)	1399.12 ± 883.69	1130 (423 to 4433)	1466 ± 952.40	1130 (361 to 4433)	0.85
Progest. (ng/dL)	0.86 ± 0.38	0.78 (0.2 to 1.75)	1.00 ± 0.89	0.81 (0.3 to 5.13)	0.90
Length of Treatment (days)	11.32 ± 1.92	11.00 (7 to 17)	10.97 ± 1.54	11.00 (8 to 15)	0.41
Injected Oocytes (*n*)	5.19 ± 2.34	5.00 (1 to 10)	7.03 ± 4.4	7.00 (1 to 17)	0.11
Fertil. Oocytes (*n*)	4.03 ± 1.72	4.00 (1 to 7)	5.81 ± 3.75	6.50 (1 to 13)	0.03
Embr. Obtained (*n*)	3.00 ± 1.26	3.00 (1 to 6)	3.94 ± 2.68	3.50 (0 to 10)	0.29
Embr. Transferred (*n*)	1.74 ± 0.82	2.00 (0 to 3)	1.64 ± 0.64	2.00 (0 to 3)	0.32
**Pregnancy Status**	**Abnormal Response**		**Normal Response**		
Yes	7 (22.6)		10 (27.80)		0.62^χ2^
No	24 (77.4)		26 (72.2)		
**Polymorphism of FSH-R**	**Abnormal Response**		**Normal Response**		
Asn/Asn Absent Polymorphism	7 (22.6)		23 (63.9)		<0.01^χ2^
Asn/Ser Heterozygote	17 (54.8)		9 (25)		
Ser/Ser Homozygote	7 (22.6)		4 (11.1)		

**Table 2 ijms-24-01080-t002:** Exclusion criteria.

Trigger Time Less than 34 h and Greater than 37 h [41,42];
Egg retrieval performed by a new doctor to avoid employee operator errors;
Errors in the administration of the trigger;
Unreachable ovaries [43];
Progesterone ≥ 1.6 ng/mL on the day of the trigger [44];
BMI ≥ 30 kg/m^2^ [45];
Cycles stimulation < 8 days [46];
PCOS [47];
Endometriosis [47];
Patients undergoing ovarian, tubal pelvic or other surgery;
Patients with endocrine disorders;
Patients with previous failure in IVF cycles;
Patients with male factor [48].

**Table 3 ijms-24-01080-t003:** Example of the equation [18] to predict the number of oocytes retrievable at the pick-up according to the age, FSH, AMH and AFC. Patient ‘ABNORMAL RESPONSE (31/67)’ showing a deficient response and Patient ‘NORMAL RESPONSE (36/67)’ showing a normal response. The response was classified as abnormal if the number of mature MII oocytes collected was 50% lower than the value predicted by the equation.

ABNORMAL RESPONDERS (31/67)	NORMAL RESPONDERS (36/67)
Age: 33	Age: 34
FSH: 7.3	FSH: 5.9
AMH: 4.76	AMH: 10.6
AFC: 14	AFC: 21
*n* oocytes retrieval (log scale) = 2.584 − 0.015 × (Age = 33) − 0.035 × (FSH = 7.3) + 0.038 × (AMH = 4.76) + 0.026 × (AFC = 14)	*n* oocytes retrieval (log scale) = 2.584 − 0.015× (Age = 34) − 0.035 × (FSH = 5.9) + 0.038 × (AMH = 10.6) + 0.026 × (AFC = 21)
EXPECTED RESPONSE: 10 MII OBTAINED RESPONSE: 3 MII 3 < 50% (10) → DEFICITARY RESPONSE	EXPECTED RESPONSE: 16 MII OBTAINED RESPONSE: 14 MII 14 ≥ 50% (14) → NORMAL RESPONSE

## Data Availability

Not applicable.

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
