# Peer review of "The Polymorphism Asn680Ser on the FSH Receptor and Abnormal Ovarian Response in Patients with Normal Values of AMH and AFC"

_ijms, 2023, doi:10.3390/ijms24021080_

Round 1
Reviewer 1 Report
read with great interest the manuscript, which falls within the aim of this Journal. In my honest opinion, the topic is interesting enough to attract the readers’ attention. Nevertheless, authors should
clarify some points and improve the discussion, as suggested below. Authors should consider the following recommendations:
In my opinion you have to improve the paper refering in the text how in both group result in a next study could be improve with inositols supplementation due multiple inositols role from ovulation induction to menopausal disorders tretament especially in pts that have tried all the known tecnique in literature as Injection of embryo culture supernatant to the endometrial cavity
Sometimes in 50% cases for male factor.
Its also important to refer hoeìw is importsnt to have in this couple a gender tailored approachand pay attention to the psychological implication.
I suggest you to read and cite some of these articles:
Myo inositol: from induction of ovulation to menopausal disorder management Myo inositol and melatonin in the menopausal transitio Injection of embryo culture supernatant to the endometrial cavity does not affect outcomes in IVF/ICSI or oocyte donation cycles: A randomized clinical trial Cryptic sperm defects may be the cause for total fertilization failure in oocyte donor cycles How does closed system vitrification of human oocytes affect the clinical outcome? A prospective, observational, cohort, noninferiority trial in an oocyte donation programAuthor Response
Please see the attachment.

Reviewer 2 Report
General comment
This study aimed to understand whether a deficient ovarian response (compared to the prediction) is correlated with amino acid 680 of the FSHr. The results of this study recommended the genetic evaluation of the FSHr to represent an accurate and predictive feature for patients undergoing assisted reproductive technology treatment. Several issues are required before reconsideration. Please view the appended comments below:
Specific comments:
1- The abstract is not organized well. Please state the context first, then the objectives, methodology, results, and conclusion portions.
2- In general, the introduction part is very long. Some sentences were not matching with the train of the hypothesis. Please try to shorten it.
3- Line 42: Anti-Mullerian hormone (AMH) levels?
4- Line 43-45: Please add more explanation for this sentence” Certainly, they have become the most applied and reliable markers in predicting the response to ovarian stimulation [3,4].
5- Line 45-47 “Hence, it has been reported that several factors including endometriosis, 45 ovarian surgery, genetic factors, smoking and most importantly the age, have an adverse 46 effect on the ovarian reserve and therefore on the AMH and AFC values [5,6]”. I think this sentence has no sense. Please reorder to match the context.
6- Line 56: the current literature or previous literature?
7- Line 89-112: The last paragraph of the introduction section should highlight the hypothesis well.
8- Line 114-115: Please state the objectives of the study to represent the content. Try to improve it.
9- Line 137: “clear association” Did the author perform correlation tests?
10- Line 146-148: why? Please state a reason.
11- Line 200-206: The conclusion section should represent the content of the study. Please be more concise.
12- Line 207: The statistical method was not presented. Please clarify in detail the statistical protocol of this study.
13- Line 209-224: please organize these sentences well to represent the groups' assignment.
14- Line 227 in table 2: please indicate the number in each group. Please address this comment to all figures and tables, too.
15- Line 232-243: Please state the ethical approval code or ID number.
16- Line 281-282: please correct some words.
17- Line 280-298: please add a reference. Also, state the kits used.
Reviewer 3 Report
This paper demonstrates an association between a function-diminishing FSH receptor (FSHR) gene variant and a poorer than expected yield of quality oocytes following ovarian hyperstimulation and IVF in mostly normal weight Italian women. The authors propose a useful mathematical equation with which to define “poor response” to ovarian hyperstimulation and IVF. They also suggest that using increased FSH doses during future IVF cycles involving such female subjects with the identified FSHR variant may generate sufficient quality oocyte yields with little risk of ovarian hyperstimulation. There are, however, crucial data missing from the methods section. Without that information, this manuscript has potentially fatal flaws in design.
p.6, 4.1, Patients. There are no data provided on the medical histories of the infertile subjects selected for this study. This information must be included in detail. Did they have accompanying endocrine abnormalities known to compromise female fertility, such as PCOS, hypothyroidism, adrenal issues, endometriosis? These would badly confound this study’s outcome. Were subjects excluded if male partners had fertility issues or medical conditions known to diminish male fertility? How many previous IVF cycles had each subject received? If so, which IVF cycle was chosen to define their inclusion in this study?
p.7, 4.3, Stimulation. What GnRH antagonist was used? What dose? What were the daily FSH injection doses? Did they vary across the IVF cycle?
Round 2
Reviewer 2 Report
Thank you very much for addressing the comments in the first round of Review. Indeed, the manuscript was improved. However, minor comments need to be addressed before the acceptance.
1- Please revise the lines of manuscript as some lines were not found (for example, between page 1 and page 2 and in the whole manuscript). The reviewer did not find some lines the authors pointed to in the AUTHOR RESPONSE File.
2- The objectives should highlight that study was doing on humans.
3- Line 627: Are you sure only 300 ul blood sample are enough?
4- Line 633-634: Please state the sensitivity, inter, and intra assay coefficients of the kits.
Author Response
Dear reviewer,
Thank you very much for the positive feedback.
Point 1: Please revise the lines of manuscript as some lines were not found (for example, between page 1 and page 2 and in the whole manuscript). The reviewer did not find some lines the authors pointed to in the AUTHOR RESPONSE File.
Response 1: The line numbers of manuscript are continuous after accepting the author revisions. There is just one gap between the line 20 and 38 (page 1): it corresponds to the line numbers of the table on the left.
Point 2: The objectives should highlight that study was doing on humans.
Response 2: Line 80 – it has been hightlighted.
Point 3: Line 627: Are you sure only 300 ul blood sample are enough?
Response 3: Yes. We followed the instructions of WizardTM Genomic DNA Purification Kit.
Point 4: Line 633-634: Please state the sensitivity, inter, and intra assay coefficients of the kits.
Response 4: Could we have more information about this question? All methodologies were performed according to the manufacturer’s instructions.
Reviewer 3 Report
The authors have responded thoughtfully to previous review issues. I have no additional concerns.
Author Response
Dear reviewer,
Thank you very much for you kind response.
Best Regards and Happy New Year.
Round 3
Reviewer 2 Report
Line 81-85: The rationale of the manuscript is still weak. The authors should improve it.
line 214- Again, please state the sensitivity, and inter and intra-assay coefficient of all measured hormones (AMH, estradiol, and progesterone) for the patients.
line 256: A blood sample of about 300 µl was collected from each patient, using a collection tube with EDTA as an anticoagulant. I understand that 300 ul may be enough for the assay but it seems very low as a sample. Please confirm again.
